# Trends and Determinants of Full Immunisation among Children Aged 12–23 Months: Analysis of Pooled Data from Mozambican Household Surveys between 1997 and 2015

**DOI:** 10.3390/ijerph20032558

**Published:** 2023-01-31

**Authors:** Marta Cassocera, Orvalho Augusto, Assucênio Chissaque, Esperança Lourenço Guimarães, Katherine Shulock, Nilsa de Deus, Maria R. O. Martins

**Affiliations:** 1Instituto Nacional de Saúde de Moçambique, Marracuene District, EN1, Bairro da Vila—Parcela n° 3943, Maputo 264, Mozambique; 2Global Health and Tropical Medicine, Instituto de Higiene e Medicina Tropical, Universidade Nova de Lisboa, Rua da Junqueira 100, 1349-008 Lisbon, Portugal; 3Faculdade de Medicina, Universidade Eduardo Mondlane, Maputo 702, Mozambique; 4Centro de Investigação em Saúde da Manhiça, Manhiça 12th Street, Distrito da Manhiça, Maputo 1929, Mozambique; 5Department of Global Health, University of Washington, Seattle, WA 98105, USA

**Keywords:** children, full immunisation coverage (FIC), trends, determinants, surveys, Mozambique

## Abstract

The 1974 Expanded Program on Immunisation has saved millions of children worldwide by promoting full immunisation coverage (FIC). However, forty years later, many sub-Saharan African countries remain well below its target of 90% FIC. This study analysed the level, trends and determinants of FIC in 4322 Mozambican children aged 12–23 months from pooled data from four national surveys between 1997 and 2015. Descriptive statistics and multivariable logistic regression models were performed to analyse the factors associated with full immunisation coverage. Overall, the coverage of fully immunised children increased from 47.9% in 1997 to 66.5% in 2015, corresponding to a 1.8% yearly increase. The needed FIC growth rate post-2015 was 4.3 times higher. Increased maternal education and a higher household wealth index were associated with higher odds of FIS. Furthermore, attending antenatal care (ANC) visits, institutional delivery and living in southern provinces were also associated with increased odds of FIS. Between 1997 and 2015, FIC among 12–23-month-old children made modest annual gains but remained well below international targets. Factors related to access to healthcare, educational level, socioeconomic status and geographical location were associated with improved FIC. Targeted interventions to expand these factors will improve immunisation coverage among Mozambican children.

## 1. Introduction

In 2015, the United Nations General Assembly established the Sustainable Development Goals (SDGs) under the slogan “leave no one behind.” The third SDG is to ensure healthy lives and promote well-being for all ages [1]. Child immunisation is one of the most cost-effective and sustainable strategies to prevent morbidity and mortality and to ensure health and well-being across the lifespan, especially among children living in low-income countries such as Mozambique [1].

Immunisation coverage is the proportion of a population vaccinated with the basic schedule during a certain period in a geographic area. It is an important indicator of the World Health Organization’s (WHO) 1974 Expanded Program on Immunisation (EPI), which aims to provide vaccines against preventable diseases to all children at risk [2,3]. Specifically, full immunisation coverage (FIC) is defined as the proportion of children who have received one dose of Bacillus Calmette–Guérin (BCG), three doses of poliomyelitis (OPV), three doses of the combined diphtheria, tetanus toxoid and pertussis (DTP or other polyvalent vaccines including DTP) and one dose of measles (either monovalent or as measles-containing vaccine combinations) between the ages of 12–23 months [4,5]. The Global Vaccine Action Plan (GVAP), a joint plan developed by UNICEF and WHO, recommends that all countries reach 90% vaccination coverage for all essential vaccines. Global FIC has increased in the last decades, contributing significantly to a reduction in morbidity and mortality from vaccine-preventable diseases [6]. For instance, from 2000 to 2016, immunisation coverage of the third dose of the DTP vaccine (DTP3) increased from 72% to 86% globally, the result of a combination of several national and international efforts [7]. However, FIC remains lower in sub-Saharan Africa (SSA), where it varies from 31–76%, far from GVAP target of at least 90% global FIC by 2020 and at least 80% FIC in every district for all vaccines in the national immunisation programme [8].

The EPI in Mozambique was established in 1979, with the ultimate goal of reducing child morbidity and mortality by vaccine-preventable diseases through immunisation services [3]. Mozambique’s child immunisation programme is provided free of charge through primary health care delivery including health facilities, mobile teams and community health workers [3]. Household surveys conducted in the country show that between 1997 and 2015, national DTP3 coverage improved from 58% to 82%, and BCG coverage reached 90% [3,9]. However, although the FIC of children aged 12–23 months increased from 47% in 1997 to 64% in 2011, by 2015, it had plateaued at 66% [9,10]. Therefore, with this declining rate, Mozambique likely did not reach the GVAP target in 2020.

In addition, Mozambique experiences substantial geographical differences in FIC, ranging from 50% in the central region to 87% in the southern region [9]. Studies conducted in different SSA countries have identified several determinants and health system factors requisite for reaching FIC. These include: institutional delivery [11,12,13]; increased maternal education [11,12,13]; residing in urban areas [11]; antenatal care (ANC) attendance [11,13]; and increased household wealth [14]. Despite the correlation between these factors and FIC, few studies have assessed the factors related to complete immunisation status in Mozambique. Those that did are restricted to the district level, leaving a large knowledge gap at the national level [15,16]. Three Demographic and Health Surveys (DHS, 1997, 2003 and 2011) and the 2015 Malaria, HIV/AIDS, and Immunisation Indicator Survey (IMASIDA) were conducted across the country between 1997 and 2015, and these provide an opportunity to measure and describe FIC and its associated determinants at the national level. The objective of this study was to analyse trends and factors associated with full immunization status among Mozambican children aged 12 to 23 months old between 1997 and 2015.

## 2. Materials and Methods

### 2.1. Study Design and Data Sources

In this study, we analysed a pooled dataset from multiple national representative cross-sectional household surveys (DHS 1997, DHS 2003 and DHS 2011, and IMASIDA 2015) [9,10,17,18]. The surveys were designed to collect characteristics of the household, including maternal and child health status, through a country-wide multistage and stratified random sample. Survey information was obtained through questionnaires administered to women who were between 15 and 49 years of age who provided information on the household’s sociodemographic, economic and health status, including maternal and child health, environment, behaviour, and pre- and postnatal care. The survey response rate of the interviewed women was high overall at 91.5%, 90.9%, 98.9% and 94.5% in 1997, 2003, 2011 and 2015, respectively. Data collection was conducted between March and June, August and December, April and November, and June and September for DHS 1997, DHS 2003, DHS 2011 and IMASIDA 2015 surveys, respectively. Detailed methodology for the survey’s participant selection has been described elsewhere [9,10,17,18]. Permission to use these datasets was granted by the Monitoring and Evaluation to Assess and Use Results Demographic and Health Survey (MEASURE-DHS).

### 2.2. Study Setting

Mozambique is a southern East African country covering 801,590 km^2^ (Figure 1). Its estimated 30,066,648 inhabitants reside primarily (65.9%) in rural areas and about 15.3% of the population are children below the age 5 years of which 17.3% are children living the second year of life (Appendix A) [19]. Following independence from Portuguese colonial rule in 1975, Mozambique’s 16-year civil war destroyed its infrastructure and displaced more than half of its population [20]. Presently, the country is classified as a low-income country (LIC) by the World Bank, with a gross domestic product per capita (average annual income) of USD 503 in 2019 [21]. According to the United Nations 2020 Human Development Index, Mozambique was ranked 181 out of 189 countries, decreasing by one point from its ranking in 2019 and 2018 [22]. The 1979 EPI launch in Mozambique introduced BCG, polio, DTP and measles vaccines to the population [3]. Gradually, more vaccines were added to the national schedule, including an updated DTP (containing additional immunogens for hepatitis B and haemophilus influenzae type B) in 2008 and pneumococcal conjugate vaccine (PCV) in 2013. Rotavirus vaccine (RV), injected polio vaccine (IPV) and the second measles dose were introduced in 2015, data from which were not captured in the 2015 survey. The measles and rubella vaccine (MRV) was introduced in 2019 [3]. Despite increased availability of these vaccines, vaccine coverage, accessibility and uptake has remained below EPI targets.

### 2.3. Survey Eligibility Criteria

The present analysis included women surveyed who had a living child aged 12–23 months at the time of the survey. Interviewers communicated with study participants using participants’ preferred language. When available, children’s health cards, which contain information on vaccination doses and related dates, were used to determine children’s immunisation status. When health cards were not available, mothers’ verbal reports by recall of the vaccines their children had received were used to determine children’s immunization status. Table 1 details when health cards and when mothers’ verbal reports were used across surveys.

### 2.4. Measurement of Variables

#### 2.4.1. Outcome Variable

For this analysis, full immunisation status is the outcome variable of interest. The WHO guideline considers a fully immunised child to have been administered eight doses in total: one dose of BCG; at least three doses of OPV (excluding that given immediately after birth); three doses of DTP; and one dose of measles vaccine between 12 and 23 months of age [23]. This outcome variable aligns with similar studies across many SSA countries [13,24,25,26,27]. Each vaccine dose variable had five response categories: (0) no vaccine; (1) vaccination dates on card; (2) reported by mother; (3) vaccination marked on card; and (8) don’t know. The categories (1), (2) and (3) were recoded as “1” to indicate those that received vaccines and (0) and (8) were recorded as “0” to indicate that no vaccination was received. The vaccination status of each of the 8 doses were then combined to compute a final variable, “immunisation status,” categorised as “1” for fully immunised and “0” for not fully immunised, capturing children who had missed one or more of the 8 doses.

#### 2.4.2. Factors Associated with Full Immunization Status

Drawn from the literature and data availability across all surveys, a total of 13 variables were selected for potential association with Mozambican children achieving full immunisation status. The variables included in this analysis were: (1) maternal age at the date of the survey, grouped as 15–24, 25–34 and ≥35 years old; (2) maternal educational level defined as illiterate, primary and secondary or above; (3) marital status, which originally had six categories (single, married, living with a partner, separated, divorced, widowed) and which was recoded into three categories (single/never in a union, married/living with a partner and divorced/separated/widowed); (4) mother’s occupation defined as unemployed (not working/household domestic) and employed (professional, clerical, sales, skilled and unskilled manual, services, agriculture, self-employed and all others); (5) geographic location whereby the provinces were included separately and aggregated to reflect the three regions (northern, central and southern) of the country; (6) area of residence, categorised as urban and rural; (7) wealth index, which was maintained as defined in each survey: poorest, poor, middle, richer and richest; (8) religion, the categorization of which varied among surveys but was regrouped to reflect four standardised responses to all datasets: Catholic, Islamic, Protestant and others, which also included no religion (although religion was analysed in all surveys, this variable was not included in the multivariable regression model due to its possible collinearity); (9) number of ANC visits (no visits, 1 to 3, ≥4); (10) place of delivery (at home/other, at health facility); (11) sex of the child (male and female); (12) birth order (1, 2 to 3, ≥4); and (13) health card possession (seen, not seen, no card).

#### 2.4.3. Statistical Analysis

Descriptive statistics were used to summarise the selected factors. The FIC and its 95% confidence interval (95CI) were computed for each survey per each potential determinant. Then, per determinant we estimated the annual exponential growth rate (AGR) of FIC, which is a relative yearly growth rate between two subsequent surveys, s and s*, using the formula:$$annual.growth.rate_{s\;to\;s*}=e^{\frac{log\left(Cov_{s*}\right)-log\left(Cov_{s}\right)}{year_{s*}-year_{s}}}$$
where Cov_s_ and Cov_s*_ represent coverages of two subsequent surveys, respectively, and year_s_ and year_s*_ are the years of the two subsequent surveys, respectively. The standard error for AGR was computed through the delta method, from which 95CI were produced. In addition to the AGR, we computed the needed annual growth rate (NAGR) to reach 90% FIC by 2020 since 2015. To improve readability, we present the AGR and the NAGR in percentages by subtracting 1 and multiplying by 100, abbreviated AGRp and NAGRp, respectively.

To analyse factors associated with full immunisation status, we used a mixed-effects logistic regression with the primary sampling unit (PSU) as random intercepts. This analysis assesses the pooled association across the period of time between 1997 and 2015. All covariates included in the model are based on the literature review and data availability. In addition, a linear term of the year of the survey was introduced in the model to capture potential changes over time. A linear term was chosen to improve the interpretability of the parameters. The province and the urban/rural variables were included as covariates in the models. The province is included as dummy indicators coded as sum contrasts. Therefore, its coefficients are to be interpreted as deviations from the overall country average. In addition, complex sampling logistic regression (reported in the Appendix A) specific to each survey was performed as a sensitivity analysis for the overall analysis. We report the odds ratio (OR), 95CI and the p-values. Data management procedures and descriptive and complex survey analysis were conducted using Statistical Package for the Social Sciences (SPSS) version 26.0 [28]. The mixed-effects model was performed in R version 4.0.3 (R Foundation for Statistical Computing, Vienna, Austria) using maximum likelihood estimation with 30 Adaptive Gaussian Quadrature’s through the package GLMMadaptative [29].

## 3. Results

### 3.1. Sociodemographic and Economic Characteristics of the Mothers and Children

We included data of 1240 (1997), 1931 (2003), 2325 (2011) and 1131 (2015) children aged 12–23 months. Across all surveys, the majority of mothers surveyed were between 25 and 34 years of age, had only obtained a primary school education level, were married or living with a partner, and belonged to the poorer or poorest wealth index. Although maternal employment decreased by 18.7% between 1997 and 2015, improvements were observed in ANC visits, which increased from 48.7% to 57.3%, and in institutional delivery, which increased from 48.9% to 72.1% between 1997 and 2015 (Table 1).

### 3.2. Full Immunisation Coverage Rate and Trends among Children

The prevalence of fully immunised children was 47.9% (95% CI: 42.8 to 52.9%) in 1997, 63.8% (95% CI: 60.6 to 66.9%) in 2003, 64.7% (95% CI: 64.7 to 67.9%) in 2011 and 66.5% (95% CI: 62.1 to 70.6%) in 2015 (Table 2). For most years, FIC was higher in children whose mothers were 15–24 years old, with 52.7% (95% CI: 46.4 to 58.2%) in 1997, 68.7% (95% CI: 64.5 to 72.6%) in 2003 and 66.7% (95% CI: 61.9 to 71.2%) in 2011. In 2015, however, the highest coverage (67.1%) was observed among children from mothers 25–34 years old (95% CI: 60.9 to 72.7%). Across all surveys, FIC was higher in children whose mothers had completed secondary school or above, were single or had never been in a union, lived in urban areas and in the southern region of the country, belonged to the highest household wealth index, attended four or more ANC visits and had institutional delivery. The child’s gender did not present significant differences across all surveys. Regarding birth order, FIC was higher among firstborn children in the 1997 to 2011 surveys. However, in 2015 the coverage was higher for second- and thirdborn children. In all surveys, possession of a health card was related to higher rates of FIC (Table 2).

### 3.3. Time Trends and Needed Growth Rate to 90% FIC by 2020

Overall, yearly, Mozambique’s FIC increased by 1.84% (AGRp: 1.84%, 95CI: 0.23; 3.48%) between 1997 and 2015 (Table 3). The fastest gain occurred between 1997 and 2003, with a yearly increase of 4.89% (AGRp: 4.89%, 95CI: 2.87 to 6.95%). This was followed by an almost flat increase of a 0.18% (AGRp: 0.18, 95CI: −0.71; 1.07) growth rate between 2003 and 2011 and a 0.69% (AGRp: 0.69%, 95CI: −1.35; 2.77%) growth rate between 2011 and 2015. This trend pattern was similar across different sociodemographic characteristics. Nevertheless, the overall fastest gains were observed among children born to a pregnancy without antenatal care (AGRp: 11.6%, 95CI: 0.79; 23.65%), and children from the poorest wealth quintiles households (for poorest AGRp: 5.7% 95CI: −2.46; 14.54%; for poorer AGRp: 3.94% 95CI: −1.13; 9.26%).

Despite an overall growth trend, the required pace (NAGRp: 6.24%, 95CI: 4.98; 7.70%) since 2015 to reach a 90% FIC by 2020 was never reached in any time segments of the analysis (Table 3).

### 3.4. Factors Related to Full Immunisation Status

The crude odds ratio and baseline descriptive statistics of each survey are shown in Appendix A for the 1997, 2003, 2011 and 2015 surveys, respectively. A total of 4322 children aged 12–23 months were included in the pooled multivariable analysis, which showed that maternal education, household wealth index, religion, ANC visits, place of delivery, birth order and geographic area were factors associated with children’s full immunisation status as presented in model 1 (Table 4).

Compared to children whose mothers had no formal education, children born to mothers who attained primary and secondary or above were 1.41 times (AOR = 1.41, 95% CI: 1.19 to 1.68) and 1.70 times, (AOR = 1.70, 95% CI: 1.24 to 2.34), respectively, more likely to achieve full immunisation status. Regarding household wealth index, children from households in the middle, richer and richest quintiles had 1.47 times (AOR = 1.47, 95% CI: 1.16–1.86), 1.43 times (AOR = 1.43, 95% CI: 1.10 to 1.87) and 1.47 times, (AOR = 1.47, 95% CI: 1.04 to 2.10), respectively, higher odds of becoming fully immunised compared to children from the poorest quintile households. Compared to children whose mothers identified as Catholic, children whose mothers identified with other religions were 24% (AOR = 0.76, 95% CI: 0.59 to 0.96) less likely to achieve complete immunisation. Children whose mothers attended 1 to 3 and ≥4 ANC visits had 4.16 (AOR = 4.16, 95% CI: 3.09 to 5.61) and 4.86 (AOR = 4.86, 95% CI: 3.61 to 6.54) higher odds, respectively, of achieving complete immunisation status compared to children whose mothers had not attended ANC visits. The analysis also showed that children whose mothers delivered in health facilities were 1.68 (AOR = 1.68, 95% CI: 1.42 to 1.98) times more likely to be completely immunised compared to those whose mothers delivered at home or elsewhere. Compared to firstborn children, the odds of being fully immunised decreased by 28% (AOR = 0.72, 95% CI: 0.55 to 0.94) among fourthborn children. Among provinces, compared to Maputo City, children from Maputo Province had 1.77 (AOR = 1.77, 95% CI: 1.10 to 2.86) times higher odds of completing the full immunisation schedule. Children from Sofala had similar (AOR = 1.00. 95% CI: 0.65 to 1.56) chances as children in Maputo City. Children from the provinces of Manica, Zambézia and Nampula were 44% (AOR = 0.56, 95% CI: 0.37–0.87), 55% (AOR = 0.45, 95% CI: 0.28 to 0.71) and 53% (AOR = 0.47, 95% CI: 0.30 to 0.76) less likely, respectively, to be fully immunised than those in Maputo City.

## 4. Discussion

This study assessed the level and trend of FIC and associated factors among children aged 12 to 23 months, based on household surveys conducted in Mozambique in 1997, 2003, 2011 and 2015. Mozambique’s FIC rose from 47.9% to 66.5% in the years between 1997 and 2015, corresponding to a cumulative yearly multiplicative growth of 1.8%. However, this level falls far below the GVAP’s recommended target of ≥90% national coverage by 2020. Furthermore, between 2011 and 2015, the FIC rose by only 0.7% yearly. Compared regionally, the country’s latest FIC in 2015 (66.5%) is the lowest among other Eastern African countries such as Malawi (76%, 2015–2016 DHS), Zambia (75%, 2018 DHS) and Tanzania (75%, 2015–2016 DHS). Among the West and Central African countries of Guinea, Mali, Ivory Coast, Togo, Angola, Cameroon and the Democratic Republic of Congo, Mozambique has the highest FIC [30,31,32,33,34,35,36,37,38,39,40]. These differences can be attributed to the sociocultural and political environments, and the health services’ performance and coverage [13,41]. The present study’s multilevel analysis showed that many factors related to the mother are directly associated with the child’s immunisation status outcome, particularly maternal education level. The household wealth index was also significantly associated with FIC. These results are aligned with studies conducted in other developing countries, such as Senegal [13], Ethiopia [27,42] and Papua New Guinea [14], and in several East African countries [43]. Some studies have hypothesised that mothers who are well-positioned economically are more likely to understand public health messages such as those related to the uptake of child immunisation services [41]. Moreover, other studies have shown that mothers who are socially at an advantage are exposed to environments that may help them to capture important health information [14,44]. Data from Mozambique’s 2017 population census estimated that the national illiteracy rate attenuated from 50.4% in 2007 to 39% in 2017. Concerningly, differences between gender demonstrate that almost half of all adult women (49.4%) are illiterate compared to men’s 27.2% illiteracy rate. Geographic location is also important; between rural and urban environments, 62.4% of rural women were found to be illiterate, compared to only 25.7% of urban women [45].

In the present study, mothers who profess to following religions other than Catholicism were associated with the lowest odds of having a child fully immunised compared to mothers professing Catholicism. This was also observed in Ghana’s 1998 to 2014 pooled DHS data, which showed that compared to children whose mothers professed Christian religions, children whose mothers professed traditional religions or who had no religion had 40% and 42% lower odds of achieving complete immunisation status, respectively [46]. Moreover, a multi-country study conducted in SSA found that in 9 out of 15 (60%) countries, FIC was associated with children with Christian mothers, while children whose mothers prescribed to folk religions had a lower probability of achieving complete immunisation [5]. Additionally, in our study, the 2015 survey showed that children whose mothers identified as Islamic had 2.62 times higher odds (Appendix A) of achieving complete vaccination when compared to children whose mothers were Catholic. This has previously been observed in Mozambique and also in Liberia, where the FIC was higher among children whose mothers identified as Muslim [5]. Mozambique’s last national census reported a slight increase in people who identify as Muslim, from 17.9% in 2007 to 18.9% in 2017, while the number of those identifying as Catholic decreased from 28.4% to 27.2%, respectively [45]. However, in the present study, the Christian religion is disaggregated into Catholic and other Protestants, unlike other studies here discussed.

Our results indicate that children born to mothers who attended one or more ANC visits were more likely to be fully immunised compared to children whose mothers did not. Similar associations with ANC visits and FIC were seen in other low to middle income (LMIC) SSA countries, including Mozambique [13,27,42,43,47,48]. According to a national assessment of 1643 Mozambican health facilities in 2018, 92% of them provided prenatal care and 90% provided routine child immunisation services. This may explain the association found between ANC visit attendance and FIC and support the idea that when mothers access prenatal care or clinics in general, they are more likely to be exposed to and uptake immunisation services [49]. Mozambique’s national health education efforts, including health weeks and the Reach Every District program, which aims to bring maternal and child health services directly to communities, may have also contributed to improved FIC by increasing the quantity of and access to ANC services [3,13,16]. Additionally, other studies show that mothers who attend ANC visits [42] are advised by health professionals to adhere to the recommended prenatal and child immunisation schedule [27], confirming that for mothers who attend ANC, decisions about vaccination begin long before delivery [50].

In the present analysis, children whose mothers delivered in health facilities were 1.68 times more likely to achieve complete immunisation. Similar results were also reported in Senegal and Ethiopia, with 1.67 and 2.10 times higher odds, respectively [13,48]. In contrast, in Nigeria, children whose mothers delivered in a health facility were found to be 38% less likely to obtain full immunisation [51]. Interestingly, this contradicts findings in other regions of the world. For example, in Myanmar, children whose mothers delivered at home were more likely to achieve full immunisation [47]. With respect to birth order, our results show that children born after the firstborn (higher birth order) experience a decrease in the probability of becoming fully immunised. This aligns with birth order and FIC results in Cameroon, Nigeria, Papua New Guinea and the Sinana District in southeast Ethiopia [14,42,51,52]. Various explanations have been raised to account for this phenomenon since it is sometimes presumed that increased maternal experience would lead to better health outcomes among children born later. Those studies suggest that children higher in the birth order may compete for resources as the household conditions may have changed compared to when older siblings were born, [52,53] leading to decreased parental ability to fulfill immunisation services for subsequent children [51].

The present study has raised important geographical differences, with full immunisation status found to be associated with children from southern provinces compared to the central and northern provinces. This aligns with two prior studies conducted at the district level. In Magude district, a southern province of Maputo, FIC was found to be 71.8% [48], while in the rural districts of Gurué and Milange, in the central province of Zambézia, FIC was much lower at 49.7% and 48.0%, respectively [16]. Although the number of health facilities has increased in Zambézia and Nampula, the ratio of maternal and child health (MCH) nurses to the population remains low [54]. In addition, these provinces’ high populations, low number of health care workers per capita and inadequate health infrastructure could contribute to their low FIC observed. Moreover, the long distances travelled by the population to the health facility, access to transportation and the dispersion of communities vary across these provinces [3]. These findings highlight the need for greater investment of the EPI in reaching the most vulnerable, regardless of the area of residence. Studies conducted elsewhere have also reported an association between geographic location and children’s immunisation status. These studies attributed this difference to the cultural beliefs of diverse patient populations, the low quality of services provided by health professionals in these areas [13] and logistical challenges such as inadequate vaccine storage cold chains [13,42]. In India, an association between geographic variation and negative health outcomes, including low FIC, has been attributed to a high poverty level in some areas, parental preferences for male children, low maternal education levels and a lack of maternal autonomy [13,55].

There are a few limitations in the present study. First, the analysis of the factors associated with FIC lacks factors not related to the mother and the household such as, for example, political and governance issues, due to data unavailability. Some of these are captured in the random-effects included in the analysis. Second, a proportion of the data used in this study’s analysis may be affected by recall bias. As noted, immunisation status of children without health cards was obtained through their mother’s verbal reports, which may be inaccurate. Although the fraction of children without a health card decreased substantially between 1997 and 2015 (from 19.7% to 8.0%, respectively), the children with unseen cards increased slightly (from 13.9% in 1997 to 17.5% in 2015) over the period of the study. Depending on the accuracy of the mothers’ recollections of their children’s vaccination history, the overall FIC could change.

## 5. Conclusions

The household survey analysis from 1997 to 2015 showed that in Mozambique, full immunisation coverage among children remains below WHO and UNICEF targets. Although the differential coverage per each factor studied did reduce over time, factors such as the maternal education level, household wealth index, ANC attendance, institutional delivery, birth order and geographic location were associated with achieving full immunisation among 12–23-month-old children. Attention to these factors in public health programming will likely help reduce coverage gaps within Mozambique.

## Figures and Tables

**Figure 1 ijerph-20-02558-f001:**
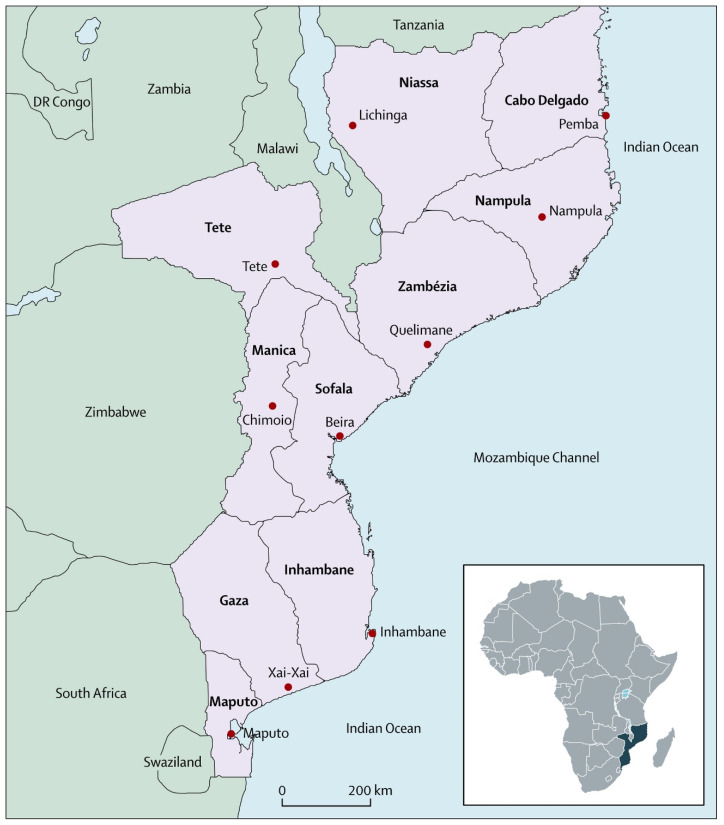
Map of Mozambique showing the province and province capitals.

**Table 1 ijerph-20-02558-t001:** Characteristics of the study population from 1997–2015 surveys.

Characteristic	1997	2003	2011	2015
	N (%)	N (%)	N (%)	N (%)
Total of children enrolled	1240 (100.0)	1931 (100.0)	2325 (100.0)	1131 (100.0)
Mother’s age group				
15–24	472 (38.0)	761 (39.4)	887 (38.1)	519 (45.9)
25–34	524 (42.3)	798 (41.3)	990 (42.6)	406 (35.9)
≥35	244 (19.7)	372 (19.3)	448 (19.3)	205 (18.1)
Mother’s education				
Illiterate	446 (35.9)	873 (45.2)	808 (34.8)	316 (28.0)
Primary	739 (59.6)	977 (50.6)	1217 (52.4)	644 (57.0)
Secondary/above	55 (4.5)	81 (4.2)	299 (12.9)	170 (15.0)
Mother’s marital status				
Single/Never in union	47 (3.8)	87 (4.5)	105 (4.5)	73 (6.5)
Married/Cohabitation	1088 (87.8)	1640 (84.9)	2000 (86.0)	871 (77.0)
Divorced/Separated/Widowed	105 (8.5)	204 (10.6)	179 (7.7)	187 (16.5)
Mother’s occupation				
Unemployed	467 (37.7)	418 (21.6)	1263 (54.5)	637 (56.4)
Employed	771 (62.3)	1513 (78.4)	1054 (45.5)	493(43.6)
Area of residence				
Urban	280 (22.6)	575 (29.7)	632 (27.2)	287 (25.4)
Rural	960 (77.4)	1357 (70.3)	1692 (72.8)	844 (74.6)
Region				
Northern	341 (27.5)	657 (34.0)	651 (28.0)	444 (39.3)
Central	473 (38.1)	772 (40.0)	1213 (52.2)	458 (40.5)
Southern	426 (34.3)	502 (26.0)	460 (19.8)	229 (20.2)
Wealth index				
Poorest	203 (16.7)	508 (26.3)	517 (22.3)	278 (24.6)
Poorer	216 (17.8)	362 (18.7)	565 (24.3)	262 (23.2)
Middle	288 (23.7)	415 (21.5)	460 (19.8)	217 (19.2)
Richer	248 (20.4)	329 (17.0)	432 (18.6)	216 (19.1)
Richest	259 (21.3)	317 (16.4)	351 (15.1)	158 (13.9)
Religion				
Catholic	378 (30.6)	541 (33.4)	703 (30.2)	334 (29.6)
Islamic	190 (15.4)	344 (21.3)	384 (16.5)	224 (19.8)
Protestant	384 (31.2)	528 (32.7)	548 (23.6)	302 (26.8)
Others	281 (22.8)	204 (12.6)	689 (29.6)	269 (23.8)
Antenatal visits				
No visits	258 (23.0)	257 (13.8)	183 (8.3)	77 (7.1)
1 to 3	317 (28.3)	564 (30.3)	926 (41.7)	382 (35.5)
≥4	545 (48.7)	1041 (55.9)	1108 (50.0)	616 (57.3)
Place of delivery				
At home/other	630 (51.1)	948 (49.1)	1069 (46.5)	316 (27.9)
Institutional	603 (48.9)	983 (50.9)	1231 (53.5)	815 (72.1)
Sex of the child				
Male	635 (51.2)	997 (51.6)	1137 (48.9)	555 (49.1)
Female	605 (48.8)	934 (48.4)	1187 (51.1)	575 (50.9)
Birth order				
1	226 (18.3)	375 (19.4)	515 (22.1)	265 (23.4)
2 to 3	472 (38.1)	649 (33.6)	779 (33.5)	410 (36.2)
≥4	542 (43.7)	907 (46.9)	1031 (44.3)	456 (40.4)
Has a health card				
No card	244 (19.7)	250 (13.0)	250 (10.8)	90 (8.0)
Yes seen	821 (66.4)	1507 (78.1)	1931 (83.1)	840 (74.5)
Yes not seen	172 (13.9)	173 (8.9)	144 (6.2)	197 (17.5)
Province				
Niassa	70 (5.6)	78 (4.0)	139 (6.0)	70 (6.2)
Cabo Delgado	70 (5.7)	169 (8.8)	188 (8.1)	104 (9.2)
Nampula	201 (16.2)	410 (21.2)	324 (13.9)	270 (23.9)
Zambezia	208 (16.8)	276 (14.3)	518 (22.3)	158 (14.0)
Tete	60 (4.8)	202 (10.4)	286 (12.3)	98 (8.7)
Manica	81 (6.5)	157 (8.1)	185 (8.0)	91 (8.0)
Sofala	124 (10.0)	138 (7.1)	224 (9.6)	112 (9.9)
Inhambane	127 (10.2)	147 (7.6)	130 (5.6)	65 (5.7)
Gaza	163 (13.2)	122 (6.3)	124 (5.3)	87 (7.7)
Maputo Province	76 (6.1)	127 (6.6)	120 (5.1)	45 (4.0)
Maputo City	60 (4.8)	106 (5.5)	87 (3.7)	32 (2.8)

Complex survey methods were used for these estimates in each survey.

**Table 2 ijerph-20-02558-t002:** Immunisation coverage in children from 12 to 23 months, 1997 to 2015 surveys.

Characteristic	Fully Immunised Children							
	1997		2003		2011		2015	
	Coverage (%)	95% CI	Coverage (%)	95% CI	Coverage (%)	95% CI	Coverage (%)	95% CI
Total of children enrolled	47.9	42.8–52.9	63.8	60.6–66.9	64.7	61.3–67.9	66.5	62.1–70.6
Mother’s age group								
15–24	52.3	46.4–58.2	68.7	64.5–72.6	66.7	61.9–71.2	66.4	61.4–73.0
25–34	45.5	38.4–52.8	60.3	55.5–65.0	65.8	61.3–70.0	67.1	60.9–72.7
≥35	44.3	31.0–58.4	61.3	55.1–67.1	58.3	52.1–64.2	65.8	56.4–74.2
Mother’s education								
Illiterate	30.7	24.8–37.4	49.1	44.4–53.9	59.1	53.6–64.4	53.5	46.3–60.5
Primary	55.1	49.4–60.6	74.1	70.6–77.4	65.6	61.7–69.4	68.0	61.2–74.1
Secondary/above	89.7	77.3–95.8	97.7	89.7–99.5	75.7	68.9–81.4	85.3	78.5–90.3
Mother’s marital status								
Single/Never in union	79.8	65.2–89.2	78.3	63.9–88.1	71.8	62.4–79.6	72.5	58.6–83.0
Married/Cohabitation	46.7	40.6–52.9	62.2	58.7–65.6	64.2	60.7–67.6	67.4	62.6–71.8
Divorced/Separated/Widowed	45.4	28.1–63.8	70.6	62.8–77.3	66.5	57.0–74.8	60.2	50.4–69.2
Mother’s occupation								
Unemployed	62.6	54.4–70.2	79.4	73.5–84.3	54.4	51.1–57.7	67.4	61.5–72.8
Employed	38.8	32.5–45.4	59.5	55.8–63.1	64.8	60.0–69.3	65.3	59.7–70.5
Area of residence								
Urban	85.2	80.5–88.9	80.7	74.8–85.4	75.4	71.2–79.1	77.9	71.0–83.5
Rural	37.0	30.3–44.2	56.7	52.8–66.3	60.7	56.4–64.8	62.7	57.4–67.7
Region								
Northern	35.5	26.8–45.2	55.0	49.1–60.8	66.9	59.7–73.5	64.7	56.4–72.2
Central	37.4	29.9–45.6	54.7	49.3–60.0	58.9	54.0–63.7	59.8	53.3–65.9
Southern	69.4	62.5–75.5	89.4	86.4–91.8	76.6	72.8–80.0	83.7	76.8–88.8
Wealth index								
Poorest	20.4	8.8–40.4	45.7	39.6–52.0	54.7	47.9–61.3	55.3	47.6–62.7
Poorer	27.4	17.7–39.9	53.9	46.6–61.1	57.8	51.9–63.5	54.9	45.1–64.4
Middle	35.7	24.6–48.6	62.1	55.7–68.1	66.6	60.0–72.5	67.1	58.0–75.1
Richer	67.8	57.3–76.7	78.7	73.7–83.1	74.2	69.4–78.4	81.0	71.4–87.9
Richest	80.4	72.8–86.3	90.9	85.8–94.2	76.2	70.4–81.2	85.1	80.0–89.1
Religion								
Catholic	42.7	34.1–51.8	61.6	55.6–67.3	61.2	55.1–67.1	59.6	51.2–67.4
Islamic	40.3	26.7–55.5	57.0	50.1–63.6	68.9	61.6–75.3	76.9	67.9–83.9
Protestant	59.8	52.3–66.9	66.3	60.7–71.4	68.1	62.4–73.3	68.6	61.3–75.2
Others	43.6	31.7–56.2	79.2	73.0–84.4	63.1	58.0–67.9	63.9	56.4–70.7
Antenatal visits								
No visits	4.0	2.3–7.0	20.4	14.1–28.4	22.9	15.9–31.7	29.0	17.4–44.2
1 to 3	53.3	44.3–62.1	62.2	56.6–67.4	68.5	64.0–72.7	60.9	53.2–68.0
≥4	62.9	56.8–68.6	74.9	71.1–78.4	68.6	64.3–72.6	73.9	68.9–78.4
Place of delivery								
At home/other	30.5	22.3–40.0	49.0	44.7–53.3	53.0	48.0–58.0	49.0	40.4–57.7
Institutional	66.3	58.9–72.9	78.1	74.3–81.5	75.2	71.6–78.6	73.3	68.6–77.5
Sex of the child								
Male	47.8	40.1–55.6	64.2	60.3–67.9	63.8	59.7–67.8	68.8	62.8–74.2
Female	48.0	38.6–57.5	63.4	58.9–67.8	65.5	61.3–69.5	64.3	58.2–70.0
Birth order								
1	57.1	48.4–65.4	73.6	68.9–78.5	68.0	62.2–73.3	67.4	59.0–74.8
2 to 3	47.1	39.9–54.4	66.8	62.2–71.2	67.8	63.0–72.2	68.8	62.1–74.7
≥4	44.7	37.8–51.7	57.6	53.0–62.0	60.7	56.3–64.9	64.0	58.2–69.4
Has a health card								
No card	0.3	0.1–0.9	5.6	2.8–11.0	9.7	6.3–14.7	11.6	5.4–23.3
Yes seen	70.3	64.7–75.4	77.6	74.2–80.7	75.8	72.6–78.8	84.0	80.0–87.3
Yes not seen	8.8	2.7–25.3	28.1	19.7–38.4	10.5	6.1–17.5	18.4	12.6–26.0
Province								
Niassa	48.7	29.8–68.1	46.6	36.8–56.6	78.1	69.5–84.8	78.0	59.1–89.7
Cabo Delgado	25.4	13.6–42.3	58.9	47.1–69.7	58.5	46.8–69.3	86.2	78.0–91.7
Nampula	34.5	21.3–50.5	55.1	45.3–64.4	67.0	54.3–77.6	52.9	41.2–64.3
Zambezia	23.2	8.3–50.3	44.9	35.9–54.2	48.2	39.7–56.8	51.3	37.9–64.5
Tete	48.0	31.3–65.2	55.4	39.1–70.6	58.9	47.3–69.7	57.2	44.9–68.7
Manica	47.0	31.3–63.3	62.2	53.0–70.6	65.1	57.4–72.1	65.8	52.0–77.4
Sofala	50.0	23.4–76.5	64.7	53.5–74.4	78.7	69.0–86.0	69.1	56.9–79.2
Inhambane	72.3	67.4–76.8	90.6	84.4–94.5	66.4	57.2–74.5	81.0	58.5–92.8
Gaza	63.3	54.1–71.7	82.3	74.4–88.1	76.3	68.3–82.9	84.8	74.9–91.3
Maputo Province	67.4	51.0–80.3	92.5	85.9–96.2	87.9	80.9–92.6	82.8	72.3–89.9
Maputo City	82.0	69.4–90.2	92.0	85.1–95.9	76.7	70.0–82.3	87.1	72.2–94.6

**Table 3 ijerph-20-02558-t003:** Yearly growth rate of full immunization coverage (FIC) and needed rate to reach 90% FIC.

Characteristic	% Annual Exponential Growth Rate				Needed Rate to Reach 90% Coverage by 2020 in %
	1997–2003	2003–2011	2011–2015	Overall 1997–2015	
Total of children enrolled	4.89 (2.87; 6.95)	0.18 (−0.71; 1.07)	0.69 (−1.35; 2.77)	1.84 (0.23; 3.48)	6.24 (4.98; 7.70)
Mother’s age group					
15–24	4.65 (2.45; 6.90)	−0.37 (−1.50; 0.78)	−0.11 (−2.90; 2.76)	1.33 (−0.64; 3.35)	6.27 (4.28; 7.95)
25–34	4.81 (1.74; 7.96)	1.10 (−0.20; 2.41)	0.49 (−2.24; 3.30)	2.18 (−0.16; 4.57)	6.05 (4.36; 8.12)
≥35	5.56 (−0.12; 11.57)	−0.63 (−2.39; 1.17)	3.07 (−1.26; 7.60)	2.22 (−1.75; 6.36)	6.46 (3.94; 9.80)
Mother’s education					
Illiterate	8.14 (4.12; 12.32)	2.34 (0.65; 4.07)	−2.46 (−6.33; 1.57)	3.13 (−0.03; 6.39)	10.96 (8.27; 14.22)
Primary	5.06 (3.12; 7.04)	−1.51 (−2.43; −0.59)	0.90 (−1.88; 3.77)	1.18 (−0.64; 3.03)	5.77 (3.96; 8.02)
Secondary/above	1.43 (−0.31; 3.21)	−3.14 (−4.22; −2.04)	3.03 (0.30; 5.84)	−0.28 (−2.04; 1.52)	1.08 (−0.07; 2.77)
Mother’s marital status					
Single/Never in union	−0.32 (−3.84; 3.33)	−1.08 (−3.48; 1.38)	0.24 (−4.85; 5.61)	−0.53 (−4.06; 3.13)	4.42 (1.63; 8.96)
Married/Cohabitation	4.89 (2.41; 7.43)	0.40 (−0.57; 1.37)	1.22 (−0.95; 3.45)	2.06 (0.20; 3.95)	5.95 (4.62; 7.53)
Divorced/Separated/Widowed	7.64 (0.29; 15.52)	−0.75 (−2.83; 1.39)	−2.46 (−7.39; 2.74)	1.58 (−3.34; 6.75)	8.38 (5.40; 12.30)
Mother’s occupation					
Unemployed	4.04 (1.57; 6.57)	−4.62 (−5.70; −3.52)	5.50 (2.80; 8.28)	0.41 (−1.58; 2.44)	5.95 (4.33; 7.91)
Employed	7.39 (4.24; 10.62)	1.07 (−0.12; 2.27)	0.19 (−2.52; 2.98)	2.93 (0.61; 5.32)	6.63 (5.01; 8.56)
Area of residence					
Urban	−0.90 (−2.24; 0.46)	−0.85 (−1.88; 0.20)	0.82 (−1.57; 3.26)	−0.50 (−2.02; 1.05)	2.93 (1.51; 4.86)
Rural	7.37 (3.42; 11.48)	0.86 (−0.90; 2.64)	0.81 (−1.86; 3.56)	2.97 (0.16; 5.87)	7.50 (5.86; 9.41)
Region					
Northern	7.57 (2.62; 12.76)	2.48 (0.59; 4.40)	−0.83 (−4.74; 3.24)	3.39 (−0.21; 7.12)	6.82 (4.51; 9.80)
Central	6.54 (2.48; 10.76)	0.93 (−0.68; 2.56)	0.38 (−2.93; 3.81)	2.64 (−0.34; 5.71)	8.52 (6.43; 11.05)
Southern	4.31 (2.61; 6.04)	−1.91 (−2.59; −1.23)	2.24 (0.08; 4.44)	1.05 (−0.42; 2.53)	1.46 (0.27; 3.22)
Wealth index					
Poorest	14.39 (0.31; 30.44)	2.27 (−0.05; 4.65)	0.27 (−4.25; 5.01)	5.70 (−2.46; 14.54)	10.23 (7.50; 13.59)
Poorer	11.94 (4.18; 20.27)	0.88 (−1.23; 3.03)	−1.28 (−6.20; 3.90)	3.94 (−1.13; 9.26)	10.39 (6.92; 14.82)
Middle	9.67 (3.34; 16.38)	0.88 (−0.84; 2.63)	0.19 (−3.72; 4.26)	3.57 (−0.57; 7.88)	6.05 (3.69; 9.18)
Richer	2.52 (−0.12; 5.22)	−0.73 (−1.79; 0.33)	2.22 (−0.76; 5.28)	0.99 (−1.18; 3.21)	2.13 (0.47; 4.74)
Richest	2.07 (0.46; 3.70)	−2.18 (−3.20; −1.15)	2.80 (0.54; 5.11)	0.32 (−1.23; 1.89)	1.13 (0.20; 2.38)
Religion					
Catholic	6.30 (2.30; 10.45)	−0.08 (−1.78; 1.65)	−0.66 (−4.76; 3.62)	1.87 (−1.33; 5.17)	8.59 (5.95; 11.94)
Islamic	5.95 (−0.66; 12.99)	2.40 (0.43; 4.41)	2.78 (−0.86; 6.56)	3.65 (−0.70; 8.20)	3.20 (1.41; 5.80)
Protestant	1.73 (−0.73; 4.26)	0.34 (−1.08; 1.78)	0.18 (−3.01; 3.49)	0.77 (−1.52; 3.10)	5.58 (3.66; 7.98)
Others	10.46 (5.15; 16.04)	−2.80 (−4.09; −1.50)	0.32 (−3.07; 3.82)	2.15 (−1.26; 5.67)	7.09 (4.95; 9.80)
Antenatal visits					
No visits	31.20 (17.55; 46.43)	1.46 (−4.61; 7.91)	6.08 (−8.33; 22.75)	11.63 (0.79; 23.65)	25.42 (15.28; 38.91)
1 to 3	2.61 (−0.59; 5.91)	1.21 (−0.14; 2.59)	−2.90 (−6.19; 0.51)	0.74 (−1.85; 3.40)	8.12 (5.77; 11.09)
≥4	2.95 (1.15; 4.79)	−1.09 (−2.05; −0.13)	1.88 (−0.35; 4.16)	0.90 (−0.70; 2.52)	4.02 (2.80; 5.49)
Place of delivery					
At home/other	8.22 (2.84; 13.88)	0.99 (−0.63; 2.63)	−1.94 (−6.77; 3.13)	2.67 (−1.29; 6.79)	12.93 (9.30; 17.37)
Institutional	2.77 (0.80; 4.77)	−0.47 (−1.28; 0.35)	−0.64 (−2.52; 1.28)	0.56 (−0.97; 2.11)	4.19 (3.04; 5.58)
Sex of the child					
Male	5.04 (2.04; 8.13)	−0.08 (−1.16; 1.01)	1.90 (−0.73; 4.61)	2.04 (−0.18; 4.32)	5.52 (3.94; 7.46)
Female	4.75 (1.12; 8.50)	0.41 (−0.77; 1.60)	−0.46 (−3.20; 2.35)	1.64 (−0.92; 4.26)	6.96 (5.15; 9.11)
Birth order					
1	4.32 (1.51; 7.21)	−0.98 (−2.28; 0.33)	−0.22 (−3.74; 3.43)	0.93 (−1.54; 3.46)	5.95 (3.77; 8.81)
2 to 3	6.00 (3.05; 9.03)	0.19 (−1.01; 1.39)	0.37 (−2.46; 3.28)	2.13 (−0.15; 4.46)	5.52 (3.80; 7.70)
≥4	4.32 (1.32; 7.40)	0.66 (−0.66; 2.00)	1.33 (−1.49; 4.23)	2.01 (−0.33; 4.41)	7.06 (5.34; 9.11)

**Table 4 ijerph-20-02558-t004:** Mixed-effect logistic regression of fully immunised children, 1997–2015.

Characteristic	OR	95% CI	*p*-Value
Total included	4322		
Mother’s age group			
15–24	1.00	-	
25–34	1.08	0.90–1.29	0.407
≥35	1.19	0.92–1.56	0.191
Mother’s education			
Illiterate	1.00	-	
Primary	1.41	1.19–1.68	<0.001
Secondary/above	1.70	1.24–2.34	0.001
Mother’s marital status			
Single/Never in union	0.90	0.66–1.23	0.520
Married/Cohabitation	1.00	-	
Divorced/Separated/Widowed	0.82	0.65–1.05	0.112
Mother’s occupation			
Unemployed	1.00	-	
Employed	0.98	0.84–1.15	0.827
Area of residence			
Urban	1.18	0.95–1.48	0.138
Rural	1.00	-	
Wealth index			
Poorest	1.00	-	
Poorer	0.93	0.74–1.18	0.561
Middle	1.47	1.16–1.86	0.001
Richer	1.43	1.10–1.87	0.009
Richest	1.47	1.04–2.10	0.031
Religion			
Catholic	1.00	-	
Islamic	1.14	0.88–1.46	0.319
Protestant	0.80	0.64–1.01	0.057
Others	0.76	0.59–0.96	0.024
Antenatal visits			
No visits	1.00	-	
1 to 3	4.16	3.09–5.61	<0.001
≥4	4.86	3.61–6.54	<0.001
Place of delivery			
At home/other	1.00	-	
Institutional	1.68	1.42–1.98	<0.001
Sex of the child			
Male	1.09	0.94–1.26	0.275
Female	1.00	-	
Birth order			
1	1.00	-	
2 to 3	0.89	0.69–1.14	0.353
≥4	0.72	0.55–0.94	0.015
Time since 1997 (years)	1.02	1.00–1.03	0.017
Province			
Niassa	0.72	0.44–1.18	0.188
Cabo Delgado	0.76	0.46–1.26	0.282
Nampula	0.47	0.30–0.76	0.002
Zambézia	0.45	0.28–0.71	<0.001
Tete	0.72	0.46–1.13	0.155
Manica	0.56	0.37–0.87	0.009
Sofala	1.00	0.65–1.56	0.982
Inhambane	1.35	0.84–2.16	0.214
Gaza	1.47	0.93–2.32	0.099
Maputo Province	1.77	1.10–2.86	0.019
Maputo City	1.00	-	
Intercept	0.34	0.19–0.62	<0.001
Random Intercept SD †	0.4556		

† The SD is in logit scale. This is for the PSU.

## Data Availability

Datasets and materials are available in the DHS program website and were granted after the purpose and objectives of the study were declared. Administrative permissions to access raw data was not required.

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
