# Peer review of "Trends and Determinants of Full Immunisation among Children Aged 12–23 Months: Analysis of Pooled Data from Mozambican Household Surveys between 1997 and 2015"

_ijerph, 2023, doi:10.3390/ijerph20032558_

Round 1

Reviewer 1 Report

Dear authors,

Below are my observations on the submitted manuscript:

Line 72: The region was incorrectly reported as it is referred to as "xxx".

Lines 83-87: I suggest that the objectives be worded in an assertive manner. As it stands, it resembles method writing.

Lines 103 to 119, referring to the study setting: I see this section as an introductory basis for research. I think it would be convenient to present, in addition to the socio-demographic data of the country, a map that locates it on the continent, as well as the division by provinces.

Reviewer 2 Report

My comments for the authors are given below, 

Line 72, please elaborate xxx region presenting 87% FIC.

Line 93, please add one-two line explaining multistage campaign of immunization during the survey period.

I can see a significant proportion of study participants were illiterate (mentioned by the authors), then how the data collection team approached them and collect these information?. by talking to them in local language?. If yes, then add this in methods section.

Please provide information about the EPI centers, their establishment and operations , if FIP is provided free of charges by the state then add this also.

I am not convinced about the data collection from mothers or availability of vaccination cards after many years?. add this in study limitations.

Time of data collection is not mentioned by the authors. How did they collect it? by visiting individual home? What was the response rate of the participating mothers? method section lacks these information.

UNICEF target of immunization is not mentioned in the manuscript so better to omit it in the conclusion.

Round 2

Reviewer 2 Report

Thank you for updating the manuscript as per my comments. Best wishes